# Shaping the Future of Immunotherapy Targets and Biomarkers in Melanoma and Non-Melanoma Cutaneous Cancers

**DOI:** 10.3390/ijms24021294

**Published:** 2023-01-09

**Authors:** Pavlina Spiliopoulou, Olga Vornicova, Sofia Genta, Anna Spreafico

**Affiliations:** 1Princess Margaret Cancer Centre, University Health Network, Toronto, ON M5G 2C1, Canada; 2Mount Sinai Hospital, University Health Network, Toronto, ON M5G 1X5, Canada

**Keywords:** melanoma and non-melanoma skin cancers, novel immune checkpoints, biomarkers of immunotherapy resistance, biomarkers of immunotherapy response, microbiome, circulating tumor DNA

## Abstract

Recent advances in treating cutaneous melanoma have resulted in impressive patient survival gains. Refinement of disease staging and accurate patient risk classification have significantly improved our prognostic knowledge and ability to accurately stratify treatment. Undoubtedly, the most important step towards optimizing patient outcomes has been the advent of cancer immunotherapy, in the form of immune checkpoint inhibition (ICI). Immunotherapy has established its cardinal role in the management of both early and late-stage melanoma. Through leveraging outcomes in melanoma, immunotherapy has also extended its benefit to other types of skin cancers. In this review, we endeavor to summarize the current role of immunotherapy in melanoma and non-melanoma skin cancers, highlight the most pertinent immunotherapy-related molecular biomarkers, and lastly, shed light on future research directions.

## 1. Introduction

The unprecedented survival gains observed with immunotherapy in cutaneous melanoma, over the course of just one decade, are demonstrated by the improvement of 1-year survival rate of 25% to more than 50% for patients with metastatic disease [1,2]. Immunotherapy aims at circumventing the immune evasion mechanisms employed by cancer cells [3], which exploit immune checkpoint receptor pathways to decrease immune activation [4,5,6,7]. Cytotoxic T lymphocyte antigen-4 (CTLA-4) and programmed cell death protein 1 (PD-1) are two critical receptors expressed on T lymphocytes that, upon ligand binding, trigger a signaling cascade that inhibits T-cell activation and limits immune stimulation [8,9]. Antibodies against these receptors (such as ipilimumab, nivolumab, pembrolizumab) prevent receptor-ligand interaction, “releasing the brake” on the immune response [10,11]. Immune checkpoint inhibitors (ICIs) that abrogate CTLA-4 and PD-1 are now the standards of care for advanced melanoma. The observed efficacy in the metastatic/advanced setting has naturally led the way to exploring the efficacy of immunotherapy in earlier stages of melanoma.

In this review, we summarize the current state of play of immunotherapy treatments in melanoma and non-melanoma skin cancers. We also present evidence underpinning the therapeutic inhibition of novel immune markers. Lastly, we also touch upon preclinical and clinical molecular signatures that can guide decision making during immunotherapy treatment.

## 2. State of Play of Immunotherapy in Melanoma and Non-Melanoma Skin Cancers (NMSC)

### 2.1. Melanoma

*Current standards in adjuvant/metastatic setting.* In advanced/metastatic melanoma, which mainly encompasses unresectable stage III and stage IV disease (excluding patients with brain metastases), the most recently mature data show that treatment with a combination of anti-CTLA-4/anti-PD-1 (ipilimumab/nivolumab) confers a median overall survival of 72 months, which is almost double that of nivolumab monotherapy (36.9 months) and certainly incomparably superior to that of ipilimumab monotherapy (19.9 months) [12]. Importantly, most patients treated with ipilimumab/nivolumab (77%) or nivolumab monotherapy (69%) remain treatment-free at 6.5 years of follow-up after treatment, demonstrating significant durability of disease response. Moreover, the benefits of combined anti-CTLA-4/anti-PD-1 inhibition were similar for patients, irrespective of their *BRAF*V600 mutational and PD-L1 expression status. The success of ICI treatment in advanced melanoma, has raised the unavoidable question of treatment duration of maintenance anti-PD-1 inhibition [13], especially for those patients who experience complete and/or partial radiological response. There is certainly compelling data that in patients with complete responses to anti-PD-1, radiological disease response persists even after treatment discontinuation, especially in those who received at least 6 months of treatment [14,15]. However, the rate of disease relapses and subsequent responses to treatment re-challenge for those who decide to discontinue treatment varies between published reports [13,15,16]; Therefore there is still controversy on the subject of elective discontinuation after achievement of radiological “no evidence of disease” and studies examining the question of continuous versus intermittent anti-PD-1 treatment are on-going [17].

Despite the practice-changing outcomes of the pivotal CheckMate 067 study, the question on ICI-activity in patients with brain metastases [stage IVd by American Joint Committee on Cancer (AJCC) 8th edition] remained; these patients were excluded from study participation at the time of protocol inception [18]. Since then, a series of studies have attempted to answer this question. Collective results suggest that treatment with anti-CTLA-4, anti-PD-1, or the combination thereof, is active in patients with intracranial metastases, offering the greatest benefit to those who are asymptomatic of the malignant infiltration at the time of presentation [19,20,21,22,23]. Intracranial responses to ipilimumab/nivolumab were found to be as high as 53.5% by Tawbi et al. in patients whose metastases has not been previously treated with brain radiotherapy [19]. Two randomized studies in patients with brain metastases treated with immunotherapy have been reported so far. Long et al. found that ipilimumab/nivolumab was significantly superior to nivolumab with a 5-year survival rate of 51% compared to 34%, in patients with asymptomatic disease [20], whereas in a study by Di Giacomo et al., ipilimumab/nivolumab conferred better survival at 4 years, compared to chemoimmunotherapy (41% vs. 10%, *p* = 0.015) [21]. Outcomes appear to be inferior for patients with established neurological symptoms secondary to their metastases and hence, these patients frequently require upfront stereotactic radiosurgery, which can be safely combined with immunotherapy [24,25].

Lastly, another critical question surrounding the combination of anti-CTLA-4/anti-PD-1 treatment continues to be the optimal ipilimumab dosing. The pivotal CheckMate-067 study that led to the approval of ipilimumab/nivolumab as first line treatment in advanced/metastatic melanoma employed the dosing of 3 mg/kg of ipilimumab and 1 mg/kg for nivolumab (IPI3 + NIVO1), during the first 4 cycles of co-administration [26]. CheckMate-511 study was the first, large-scale randomized study exploring a lower dose of ipilimumab (1 mg/kg), alongside a higher dose of nivolumab (3 mg/kg), IPI1 + NIVO3, in an attempt to compare this combination to the approved IPI3 + NIVO1 dosing [27]. Although not powered enough to prove efficacy, the study demonstrated that, descriptively, efficacy measures such as overall response rate (ORR) response, progression-free survival (PFS), and overall survival (OS) were similar between groups, but importantly, toxicity was significantly lower with the IPI1 + NIVO3 regimen [27,28]. Longer follow-up data may be required to prove the non-inferiority of the lower ipilimumab dose regimen and potentially a change in clinical practice, to spare patients from the high rate of immune-related adverse events.

Attempts to augment ICI efficacy in advanced/metastatic melanoma have been made with the addition of targeted treatment in patients who have actionable mutations [29,30,31,32]. Approximately 50% of patients with metastatic melanoma harbor somatic mutations in the B-Raf proto-oncogene *(BRAF*) [33], which leads to constitutive activation of the BRAF oncoprotein [34,35]. Currently, targeted treatment with combined BRAF/MEK inhibition can be used in clinical practice for those patients with aberrantly activated signaling pathways downstream of the BRAF protein [36,37]. Concomitant treatment of BRAF/MEK inhibitors (vemurafenib/cobimetinib) with the anti-PD-L1 antibody, atezolizumab, gained approval by the US Food and Drug Administration (FDA) agency in 2020, based on the results of IMspire150 study [31]. In this study, the addition of atezolizumab to targeted therapy led to a significantly increased progression-free survival (PFS) in patients with somatic *BRAF*V600 mutation-positive advanced melanoma, when compared to treatment with targeted treatment alone. Further maturing of the data from IMspire150 showed a continuous PFS benefit, however, median survival gains have not reached statistical significance and hence, the combination of targeted and ICI treatment has not been met with excitement, so far.

Additional studies combining targeted treatment with anti-PD-1 based immunotherapy (either pembrolizumab or spartalizumab), only yielded modest benefits from the combination, at the expense of heightened toxicity [29,30]. Overall, given the limited number of available systemic treatments for metastatic melanoma, the hypothesis of combining all active treatments into one line has now been replaced with questions on the optimal sequencing of these available therapies [38,39,40]. Indeed, recently published data suggest that for patients with actionable *BRAF* mutations, prioritizing treatment with ICI as first line of treatment, followed by targeting BRAF/MEK only upon disease progression leads to survival gains for this subset of melanoma patients [38,39].

With the rapid pace of developments in the therapeutic field of melanoma, prior immunostimulatory treatments such as inteleukin-2 (IL-2) were superseded by immune checkpoint inhibitors. Although overall response rates with systemic IL-2 are modest (<20%), patients achieving complete response with IL-2 tend to experience remarkably durable benefits and high rates of long-term survival, sometimes exceeding the 10-year mark [41,42,43,44]. Interestingly, a retrospective comparative study found that response rates to high-dose IL-2 are greater among patients treated with prior ipilimumab compared to patients with no prior exposure to ICI therapy (ORR 21% vs. 12%) [45].

*Adjuvant/neo-adjuvant setting.* For high-risk, early-stage melanoma, the benefit of adjuvant immunotherapy was underpinned by a seminal study by Eggermont et al. in which the CTLA-4 inhibitor ipilimumab was found to increase the 3-year recurrence free rate by more than 10% ((46·5% (95% CI 41.5–51.3) vs. 34.8% (30.1–39.5)) compared to placebo [46]. This study included stage IIIa/b/c without in-transit metastases, based on the former AJCC staging version (7th edition), whereby the IIIa subgroup included patients with thicker melanoma and of higher risk prognosis compared to the newer AJCC 8th edition. Importantly, this study also highlighted the need to further define the risk-benefit ratio of adjuvant immunotherapy in the face of unacceptably high toxicity rate with adjuvant anti-CTLA-4 treatment, which was administered at 10 mg/kg, a higher than usual dose. Drug-related deaths due to colitis and myocarditis were observed at this dose of ipilimumab and since then, subsequent studies focused on examining single-agent immunotherapy with anti-PD-1 inhibitors following definitive surgery. Instead of adopting anti-CTLA-4 as adjuvant treatment, anti-PD-1 treatment with nivolumab [47] and later on with pembrolizumab [48], were established as effective strategies against disease recurrence, with improved toxicity profiles compared to ipilimumab. It is worth noting that on the one hand, nivolumab was tested against high dose ipilimumab in stage IIIb/IIIc/IV (AJCC 7th edition) population of patients who underwent complete tumor resection. On the other hand, pembrolizumab was tested against a placebo and in the stage III patient population that also included stage IIIa melanoma (with a lymph node metastasis of more than 1 mm in dimension), a sub-cohort of patients not included in the study of adjuvant nivolumab versus ipilimumab [47].

An exciting new step forward for early-stage melanoma was the recent FDA approval of pembrolizumab as adjuvant therapy for resected stage IIb and IIc melanoma (in accordance with 8th AJCC edition).This is melanoma without lymphatic dissemination which is either between 2–4 mm thick and ulcerated, or thicker than 4 mm irrespective of ulceration. The approval was based on an improvement in recurrence-free survival in KEYNOTE-716 study with a 35% reduction in risk of recurrence with pembrolizumab, compared to placebo (hazard ratio (HR) 0.65 [95% CI 0.46–0.92]; *p* = 0.0066) [49].

Aside from the adjuvant/metastatic settings, current research in the sphere of neo-adjuvant immunotherapy is expected to soon change the treatment paradigm for early-stage high-risk melanoma. Based on the hypothesis that the presence of tumor (and its antigenicity), as well as the presence of peritumoral lymphocytes, will support neoadjuvant ICI mediate a robust anti-tumor response before tumor resection, neo-adjuvant ICI is being examined in a series of studies [50,51,52]. From a clinical standpoint, ICI administration before surgery can (a) reduce tumor burden and hence surgical morbidity, (b) determine therapy efficacy in an individual patient basis for possible additional adjuvant therapy, and (c) use the pathological response as a surrogate biomarker of ICI efficacy and event-free survival. Initial reports from the OpACIN study showed that only two cycles of neo-adjuvant ICI (with a combination of ipilimumab/nivolumab) can augment the expansion of T cell clones and upregulate favorable immune gene signatures that lead to pathological complete responses in the tumor microenvironment [50]. At the ICI doses used in this study; however, the toxicity rate was considerably high and superiority over adjuvant approach was not established. Subsequently, in the OpACIN-neo study which examined a lower dose of ipilimumab and a higher dose of nivolumab, pathological responses were found to translate into durable clinical benefit with 97% of patients with complete responses being recurrence-free at 2 years following diagnosis [52]. In the most recently reported SWOG S1801 randomized study of neo-adjuvant anti-PD-1 blockade, followed by surgery and further adjuvant anti-PD-1, versus adjuvant anti-PD-1 alone in stage III melanoma, recurrence-free survival was 72% at 2 years with the neo-adjuvant approach compared to 49% with adjuvant only treatment [53]. This recent data is thought compelling enough to change the treatment paradigm of early-stage melanoma and crystalize the role of immunotherapy in the neo-adjuvant space.

Another interesting approach into maximizing the anti-cancer immune response in the neo-adjuvant setting, by capitalizing on the presence of the tumor and its antigenicity, is the addition of intralesional administration of oncolytic virus directly into the immune microenvironment. Talimogene laherparepvec (T-VEC) is a genetically modified herpes simplex type 1 virus that selectively infects and replicates tumor cells, whilst expressing granulocyte macrophage colony-stimulating factor (GM-CSF) which, in turn, stimulates antigen-presenting cells to initiate and propagate lymphocyte activation. T-VEC is currently used as monotherapy in Europe, US, and Australia for the treatment of unresectable IIIb, IIIc, and IV M1a stage melanoma (no visceral metastases), based on the results of the OPTiM study, which showed a 19% durable response rate (≥6 months) and a median OS of 23.3 months, at its final analysis [54]. Despite the fact that the addition of T-VEC to standard of care ICI with Pembrolizumab was not found to elicit a synergistic effect in ICI-naïve patients with stage IIIb-IVM1c melanoma [55], there still remains the hypothesis that T-VEC can synergize with systemic ICI in the neo-adjuvant setting to achieve local clearance and increase recurrence-free survival following surgical resection. The results of the recently registered NIVEC [56] trial, whereby nivolumab will be combined with T-VEC pre-operatively, will be greatly anticipated.

*New promising treatments/future standards.* Unsurprisingly, given the success of immune checkpoint inhibitors in melanoma, current research directions are focusing on the discovery of novel immune checkpoint molecules that could augment the efficacy of ICI in melanoma even further, or reverse the emergence of resistance to the available immune checkpoint blockers.

Lymphocyte-activation gene 3 (LAG-3, also known as CD223), is a cell membrane protein that is upregulated in activated T cells and binds major histocompatibility complex (MHC) class II molecules with high affinity [57,58]. LAG-3 is overexpressed on the membrane of tumor infiltrating lymphocytes and inhibition of LAG-3 results in a more efficient immune-mediated tumor clearance [59,60,61]. The randomized RELATIVITY-047 study which compared the co-formulation of anti-PD-1/anti-LAG-3 antibody, opdualag, versus anti-PD-1 treatment only in patients with untreated advanced melanoma, confirmed that abrogation of these two immune checkpoints is superior to nivolumab alone, with a PFS that is comparable to the combination treatment of anti-CTLA-4/anti-PD-1; although these two combinations have not been directly compared as yet in the same study [62]. Given that the toxicity profile of anti-PD-1/anti-LAG-3 dual inhibition appears more favorable than that of anti-CTLA-4/anti-PD-1, there is a high expectation for this new ICI combination to be soon compared with the standard-of-care anti-CTLA-4/anti-PD-1, in a randomized fashion in advanced melanoma. Anti-LAG-3 inhibition is also being examined in the neo-adjuvant space, almost in parallel with its development in advanced disease, and has already demonstrated exciting results. In a multi-arm study examining different neo-adjuvant regimens (NCT02519322), in patients with resectable clinical stage III or oligometastatic stage IV melanoma, neoadjuvant relatlimab with nivolumab resulted in high pathological complete response (pCR) rate (57%; 95% CI, 37–75%) and improvement in the 2-year recurrence-free survival (RFS) rate in patients who achieved any degree of pathologic response compared to those without a pathologic response (*p* = 0.005). Neoadjuvant anti-LAG-3/anti-PD-1 treatment mediated tumor infiltration by memory CD4^+^ and effector CD8^+^ T cells in the post-treatment tumor specimens of patients with favorable treatment response. Overall, these findings indicate that LAG-3 is now established as the third targetable immune checkpoint in melanoma.

Other immune checkpoint molecules exhibiting signals of activity in melanoma are (i) the T cell immunoreceptor with immunoglobulin and ITIM domain (TIGIT) and (ii) the glucocorticoid-induced TNF receptor (GITR). TIGIT down-regulate T cell and NK cell functions upon binding to its two ligands, CD155 and CD112, which are expressed by both tumor cells and antigen presenting cells [63]. When TIGIT inhibition was combined with anti-PD-1 inhibition in vitro, it was found to enhance tumor antigen specific CD8^+^ T cell expansion and activity, as compared with anti-PD-1 inhibition alone [64]. Double inhibition also increases proliferation and function of tumor infiltrating lymphocytes (TILs) isolated from patients with melanoma. Studies currently investigating TIGIT inhibition in patients with advanced melanoma are depicted in Table 1. Another target of interest is the glucocorticoid-induced TNF receptor (GITR), a member of the tumor necrosis factor (TNF) receptor superfamily. GITR mediates immunosuppression through constitutive expression on T regulatory cells (Tregs), whereas its expression is upregulated on CD4^+^, CD8^+^ T cells, and natural killer (NK) cells, only upon T-cell activation [65,66,67]. Hence, targeting the GITR pathway promotes antitumor activity of T cells through T cell proliferation and enhanced effector function, abrogation of Treg function, as well as by upregulating IL-2 and IFNγ pathways [68,69,70]. There are several GITR inhibitors in clinical development and MK-4166, a humanized IgG1 antibody against GITR showed promising response rates when combined with anti-PD-1 in ICI naïve patients; it did not however exhibit activity in patients with prior exposure to ICI [71].

The critical question needing urgent answer in the field of immune-oncology (IO) in advanced melanoma is undoubtedly that of reversing the emergence of ICI-resistance and discovering novel therapeutic options for patients harboring ICI-resistant disease. Ample pre-clinical research is currently focusing on answering this question and we endeavor to address some of it later in this review. From a clinical standpoint, promising results were recently reported by the LEAP-004 study [72]. In this single-arm phase 2 study, patients who were exposed and whose disease had progressed on anti-PD-1/PD-L1-based immunotherapy, were treated with the combination of pembrolizumab plus lenvatinib. Lenvatinib, a tyrosine kinase inhibitor that selectively inhibits VEGFRs 1–3, FGFRs 1–4, platelet-derived growth factor receptor α, RET, and KIT [73,74], was found to enhance the anti-tumor response of anti-PD-1 blockade in preclinical models [75,76]. In LEAP-004 study, up to 21% of patients experienced disease response, amongst them complete and partial responses [72], supporting the hypothesis that abrogation of pivotal intracellular pathways can reverse tumor resistance to anti-PD-1/PD-L1-based treatments. This hypothesis has generated several combinatorial clinical studies in patients with ICI-resistant melanoma (Table 1).

Immense excitement in the sphere of ICI-resistant melanoma has also been sparked by the efficacy of adoptive cell therapy, in the form of treatment of surgically harvested tumor-infiltrating leucocytes (TILs). Lifileucel (LN-144) is an autologous TIL product that utilizes harvested tumor-tissue T cells, using a centralized manufacturing process. Given that melanoma is a disease characterized by high mutational burden, a unique cellular product that harbors polyclonal cells with diverse antigen specificity is capable of offering a tailored immune response for each patient. In a study reported by Sarnaik et al., lifileucel was administered to patients with unresectable or metastatic melanoma, previously progressed through anti-PD-1 blockade (and targeted treatment if they had *BRAF V600* mutation-positive disease) [77]. With an overall response rate of 36% and disease control rate of 80%, these results indicate that once the practices of administering cellular therapies can be streamlined, adoptive cellular therapy will undoubtedly become a standard of care for melanoma and a robust option, at least for patients with ICI-resistant disease and available resectable tumor present.

*Non-cutaneous melanoma, current, and future therapeutic pathways*. Non-cutaneous melanoma including uveal and mucosal melanoma are rare tumors, characterized by specific epidemiology, biological behavior, and molecular profile. Uveal melanoma represents only 5% of all cases of melanoma; however, it is the most common primary intra-ocular cancer [78]. While cutaneous melanoma is often characterized by *BRAF*, *NRAS* and *KIT* mutations, uveal melanoma typically harbors *GNAQ*/*GNA11*, *EIF1AX*, *SF3B1*, *BAP1* mutations and chromosomal abnormalities [79]. In more than 90% of the cases uveal melanoma is diagnosed at an early stage, when still amenable to local treatment [80]. Unfortunately, the proportion of patients who develop metastatic disease ranges from 20% for stage I to 70% for stage III [81,82]. The liver is the most frequent site of metastasis and when feasible, liver-directed treatments including surgery, radio-ablation and chemoembolization represent the treatment of choice for metastatic disease [83]. For patients with metastatic disease that is not amenable to local treatment, immunotherapy currently represent the mainstay of therapeutic options. Immune checkpoint inhibitors targeting CTLA-4 and the PD-1/PDL-1 axis have demonstrated antitumor activity in uveal melanoma. However, when compared to cutaneous melanoma, these agents have limited efficacy, resulting in an ORR between 10 and 20% when used in combination [84,85,86,87] and even lower when used as single agents [88,89,90]. Anatomical, genomic, and immunological factors are known to contribute to ICI resistance in uveal melanoma. The eye is an immune privileged organ characterized by multiple local and systemic mechanisms limiting and preventing inflammation to protect visual function. The immunotolerant microenvironment of the eye, enriched on Tregs, may impair the efficacy of ICI on the primary tumor [91]. As compared with its cutaneous counterpart uveal melanoma is characterized by lower molecular burden, with a mutation rate of respectively 0.5 mutations per megabase (Mb) for uveal melanoma, as opposed to 49.2 mutations per Mb for cutaneous melanoma [92,93,94]. Another molecular feature responsible for the lack of responsiveness to immunotherapy is the low PD-L1 expression [95,96,97]. Epigenetic changes downregulating PD-L1 and other key immune checkpoints, including LAG-3 and CTLA-4, have been observed in uveal melanoma with *BAP1* disruption [98]. These observations supported the development of novel agents exploiting alternative targets and the investigation of combinatorial strategies to overcome the resistance to anti-PD-1 and anti-CTLA-4 agents.

Tebentafusp, a bispecific gp100 peptide-HLA-directed CD3^+^ T lymphocyte engager has been recently approved by FDA for the treatment of patients with metastatic uveal melanoma, who harbor the HLA-A*02:01 allele. This approval followed the results of a randomized phase III trial in which tebentafusp resulted in prolonged overall and progression-free survival in comparison to investigators’ therapeutic choice (including dacarbazine, anti-CTLA-4, or anti-PD-1 treatment) [99]. At 1 year, the overall survival of patients receiving tebentafusp was 73% vs. 59% in the control group (HR for death 0.51; *p* < 0.001), while progression-free survival at 6 months was 31% vs. 19% (HR for disease progression 0.73; *p* = 0.01).

Different therapeutic strategies are under investigation to enhance the effect of immune checkpoint inhibitors. The histone deacetylase inhibitor entinostat was used in combination with pembrolizumab in twenty-nine uveal melanoma patients resulting in an ORR of 14%, PFS of 2.1 months, and OS of 13.4 months [100]. The clinical benefit correlated with *BAP1* status with a higher chance of response and prolonged survival in patients with *BAP1* wild type. Clinical trials are ongoing testing the efficacy of other combinatorial regimens, such as with anti-angiogenic agents (NCT05282901, NCT05308901), anti-LAG-3 (NCT04552223), TLR9 agonists (NCT04935229), PARP inhibitors (NCT05524935), and liver-directed radio-embolization (NCT02913417) (Table 2).

Mucosal melanoma, another rare subset of melanoma, accounts for 1 to 3% of all melanoma diagnoses. It can originate from the gastrointestinal, genitourinary, and respiratory tracts; however, 80% of mucosal melanomas primaries arise from the head and neck area [101,102]. Mucosal melanoma has a peculiar genomic profile characterized by higher rate of *SF3B1* and *KIT* and lower incidence of *BRAF* and *NRAS* mutations [103]. Similarly, to uveal melanoma, mucosal melanoma is characterized by low PD-L1 expression and tumor mutational burden that blunt the efficacy of ICIs [104,105]. Anti-PD-1 antibodies alone or in combination with anti-CTLA-4 represents the main treatment option; however, clinical data indicate that these agents’ efficacy is modest when compared to cutaneous melanoma [106,107]. Similarly to the other melanoma subtypes, the combination of anti-PD-1 and anti-CTLA-4 seems to improve patients’ outcomes over monotherapy. Seventy-nine patients with mucosal melanoma were included in the CheckMate-067 study and randomized to the combination of ipilimumab/nivolumab, nivolumab or ipilimumab monotherapies. In this population the combination resulted in a superior ORR (43% vs. 30% with nivolumab and 7% with ipilimumab monotherapy) and improved 5-year PFS and OS rates (5-year PFS rate: 29% vs. 14% vs. 0% and 5-year OS rate 36% vs. 17% vs. 7%) [108]. Vascular endothelial growth factor (VEGF) upregulation is a known biomarker of poor prognosis in mucosal melanoma [109] and clinical data support the combination of antiangiogenic and immunotherapeutic agents in this population. A phase 1 study testing anti-PD-1 blockade in combination with the VEGF inhibitor axitinib in 33 mucosal melanoma patients resulted in an ORR of 48% and a median PFS and OS of 7.5 and 20.7 months, respectively [110,111]. Interestingly, the biomarker analysis revealed no correlation between PD-L1 expression or tumor mutational burden and survival outcomes while an association was observed between improved PFS and a gene-expression profile signature including angiogenesis and immune-related genes [111]. The potential role of anti-VEGF inhibition and ICI in mucosa melanoma is supported also by the results of a phase II study testing atezolizumab and bevacizumab [112]. Forty-three mucosal melanoma patients were treated with this combination, achieving an ORR of 43%, with a median PFS of 8.2 months. Further agents currently under evaluation in clinical trials testing combination strategies with ICIs (Table 2).

### 2.2. Non-Melanoma Skin Cancers Overview

Non-melanoma skin cancers (NMSCs) consist of a heterogeneous group of tumors. It was not until recently that immunotherapy treatment was incorporated into the therapeutic landscape of NMSCs. In this review, we mostly discuss the most common types such as cutaneous squamous cell carcinoma (cSCC), Merkel cell carcinoma (MCC), and basal cell carcinoma (BCC). High curability with surgery and/or radiation results in a proportionally low death rate, however, due to very high incidence, accounting for 30% of all diagnosed cancer types, the absolute mortality in advanced stages is comparable to melanoma [113]. NMSCs harbor features predictive of response to immunotherapy, including high tumor mutational burden (TMB) associated with chronic ultraviolet radiation exposure, and other common risk factors like immunosuppression, as well as viral etiology (Merkel cell polyomavirus in MCC) and advanced patients age [114,115,116]. In the past decade, multiple clinical trials confirmed the efficacy of ICIs in NMSCs, mainly in SCC and MCC [117,118,119,120,121]. Moreover, more recently, immunotherapy exhibited clinically meaningful antitumor activity in advanced BCC following progression on first-line hedgehog inhibitor therapy (HHI) [122].

*Cutaneous SCC (cSCC):* Use of single-agent anti-PD-1 inhibition with either pembrolizumab or cemiplimab as an upfront first-line therapy for advanced or metastatic cSCC, that is not amenable to surgery or radiation, showed ORR of 42% and 52% in recurrent/metastatic disease and locally advanced disease, respectively [117,118]. Both drugs demonstrated durable responses [median duration of response (DOR) not reached] and extended OS (median not reached), along with tolerable toxicity profile. Favorable toxicity profile is very pertinent to this patient population, given the advanced age and high comorbidity prevalence associated with it. These practice-changing results led to the introduction of immunotherapy as a new 1st line standard of care treatment in patients with advanced SCC [123]. Following encouraging results in advanced disease, multiple studies are currently investigating the role of ICI in adjuvant and neoadjuvant/perioperative settings (Table 3). Indeed, Gross et al. recently reported results of neoadjuvant cemiplimab in stage II to IV cSCC. The study demonstrated a 68% objective response rate, as well as a 51% pathological complete response rate, and a 13% pathological major response rate [124].

*Merkle Cell Carcinoma (MCC):* MCC is a rare and aggressive malignancy with historical 5-year OS rate of 14% for metastatic disease and 27% for clinically detected locally advanced disease (nodal involvement) [125]. Similarly to cSCC, MCC is a highly immunogenic disease [114,116], representing a very good target for ICI-based immunotherapy. JAVELIN Merkel 200 trial demonstrated ORR of 33.0% and a median duration of response of 40.5 months with avelumab for pre-treated patients with advanced MCC. Reported median OS was 12.6 months [126]. Subsequently, pembrolizumab and avelumab were separately investigated in first line treatment settings, with ORR of 56–62% and complete response rate of up to 24% [120,121]. Median duration of response and median OS are not reported yet, whilst the Kaplan Meier estimate of median OS for 1st line pembrolizumab at 3 years was 59.4% for all patients, and 89.5% for responders [127]. Multiple trials are currently on-going to investigate efficacy of ICI as monotherapy or in combination with targeted agents in neoadjuvant and adjuvant settings, along with an attempt to overcome primary and acquired resistance to ICI in advanced MCC (Table 3).

*Basal cell carcinoma (BCC):* BCCs are at least twice more common than cSCC. Compared to cSCC or MCC, BCCs are much less likely to metastasize, with a reported metastatic rate of less than 0.1% [128]. Locally advanced disease is more common, usually causing significant destruction and deformation of the underlying tissues, including soft tissue, bone, and cartilage [129]. Patients with recurrent and destructive infiltration of the surrounding tissues most commonly require systemic treatment. Among published studies, Stratigos et al. recently reported that cemiplimab can induce a significant antitumor response in patients with locally advanced BCC, after progression on HHI therapy. The study reports 31% ORR with the survival data still being immature. Surprisingly, the rate of grade 3–4 adverse events was quite significant, reaching 48%, which might become a serious challenge in this vulnerable patient population [122]. Similarly to MCC and cSCC, multiple trials are ongoing for the evaluation of the ICI role in other treatment settings (Table 3).

The approval of immunotherapy-based treatment for both melanoma and NMSCs in a timeline fashion is depicted in Figure 1.

## 3. Biomarkers of Immunotherapy Response/Resistance in Melanoma

Melanoma has immensely contributed to our understanding of the mechanisms of response or resistance to ICI. However, unlike other tumor types such as lung or head & neck cancer, where PD-L1 expression by immunohistochemistry (IHC) is used to guide patient stratification to ICI treatment, PD-L1 expression by IHC has not been proven to be a reliable predictive biomarker in melanoma, as evidenced by exploratory analyses of large scale studies [130,131,132]. Tumor mutational burden (TMB), on the other hand, is a universally accepted surrogate marker of ICI response in a tumor-agnostic fashion. Melanoma has inherently a high level of TMB, showing superior responses to ICI depending on the magnitude of the TMB [133,134] but notwithstanding this, it is not routinely utilized in the clinic as a predictor of ICI response. Given the extremely favorable response of melanoma to ICI, a biomarker of ICI-resistance is perhaps more critical in the clinical management of melanoma, rather than a biomarker of response. As we gain more insight into multi-dimensional data on intratumoral, immunological, and systemic factors modulating the anti-cancer immune response, more and more host intrinsic as well as extrinsic players are starting to emerge (Figure 2).

*Genomic markers.* Loss of PTEN function has been found to have a negative correlation with response to immunotherapy. Both in preclinical work but also in patient samples interrogated through the Cancer Genome Atlas Program (TCGA), PTEN loss correlates with decreased T-cell infiltration at tumor sites and increased expression of immunosuppressive cytokines [135,136,137]. Whole exome analysis of the resistant tumor clones identified loss-of-function mutations in the genes encoding interferon-receptor-associated Janus kinase 1 (JAK1) or Janus kinase 2 (JAK2), which leads to insensitivity to INFα, β and γ [138,139]. Activating mutations in the Wnt/b-catenin pathway can also induce resistance to ICI through altering the expression of PD-L1 and PD-L2 in a broad group of tumors [140,141] and mechanistic studies specifically in melanoma have revealed that Wnt-induced decrease in the expression of the chemokine CCL4 hindered the recruitment of CD103^+^ DCs and T cells to the tumor microenvironment [142].

DNA mismatch repair deficiencies (dMMR) and consequently microsatellite instability (MSI) predispose tumor cells to the accumulation of somatic mutations and increased TMB [143]. The connection between novel somatic mutations and the generation of neo-antigens is extremely intriguing and nuanced, at the same time. In melanoma specifically, it has been demonstrated that the generation and landscape of clonal neoantigens can determine T cell infiltration and durable clinical benefit for patient [144]. Moreover, large genomic aberrations, such as somatic copy number alterations (SCNAs), have also been found to mediate immune evasion [145]. Loss of 9p21.3 locus in a pan-cancer cohort, including patients with melanoma is linked to reduced ICI responsiveness and poorer clinical outcomes [146,147]. Specifically in melanoma, high dosage of arm- or chromosome-level SCNAs were associated with poorer response to anti-CTLA-4 in a retrospective analysis [148], one such example being gains in chromosome 7 that are accompanied by poor lymphocyte infiltrate and aberrant neutrophil activation [149].

*Transcriptomic markers.* Transcriptomic readouts have also been used in an attempt to predict responsiveness to immunotherapeutic modification with ICI treatment. These mainly pertain to immune gene signatures and characterize the immune tumor microenvironment more accurately. Melanoma can be therefore stratified into two different groups: a T-cell specific, antigen presentation-related, and IFNγ signaling–related signature helps identify the immune “hot” tumors [150,151,152]. On the contrary, signatures encompassing markers related to higher prevalence of myeloid-derived suppressive cells (MDSC), tumor-associated macrophages (TAMs), or cancer-associated fibroblasts (CAFs) associated genes are more specific to immune “cold” tumors, and correlate with reduced cytotoxic lymphocyte activity [139,153,154].

Circulating markers of inflammation have also been interrogated to discover links between biomarkers and response to ICI in melanoma. Recently, Rossi et al. demonstrated that the levels of interleukin-6 (IL-6), hepatocyte growth factor (HGF), and monocyte chemotactic protein 2 (MCP-2), found in the serum of patients with melanoma whose disease did not respond to ICI, are higher compared to ICI-responders [155]. The three chemokines could be clustered in a signature with an overall negative effect on ICI response. Transforming growth factor-β (TGFβ) is a cytokine that plays important roles in angiogenesis and immunosuppression by stimulating Tregs, promoting tumor immune escape and immunotherapy resistance [156]. High expression of TGFβ1 in tumor frequently correlates with inferior survival, even when cytotoxic T lymphocyte infiltration is rich [153,154].

Recent studies also shed light on the role of extracellular vesicles, such as exosomes and microvesicles, in their ability to carry molecules that can modulate the immune response systemically [157,158,159]. Chen et al. showed that metastatic melanomas release exosomes, that carry PD-L1 on their surface and suppress CD8^+^ T cell effector function and consequently facilitate tumor growth [160]. In patients with metastatic melanoma, treated with pembrolizumab, the pre-treatment level of circulating exosomal PD-L1 by enzyme-linked immunosorbent assay (ELISA) was significantly higher in patients whose disease failed to respond to treatment.

*Immunological biomarkers in melanoma.* The tumor microenvironment plays a critical role in the anti-cancer immune response following IO treatment [161]. In metastatic melanoma, the co-occurrence of tumor-associated CD8^+^ T lymphocytes and CD20^+^ B cells has predictive value in the treatment outcome of patients receiving ICIs [162,163,164]. On the other hand, tumor infiltration by Tregs, TAMs, and MDSCs, as well as accumulation of cancer-associated fibroblasts is commonly associated with immune “cold” tumors and adoptive resistance to CPI [165,166,167]. Organized aggregates of T, B, and dendritic cells form tertiary lymphoid structures (TLS) that participate in the adaptive antitumor immune response. B cell infiltrate and the presence of TLS in tumor samples in melanoma patients were found to be predictive of response to treatment and associated with improved survival [168,169,170,171,172].

During interrogation of immune gene signatures in RNA sequencing (RNA-seq) data of baseline and on-treatment tumor samples, enrichment of a B cell signature in responders vs. non-responders at baseline and early on-treatment was observed [168]. Furthermore, tumors with low infiltration of B cells had a significantly increased risk of death in comparison to the B-cell-high signature cohort of patients [169]. The density of CD20^+^ B cells and TLSs and the ratio of TLSs to tumor area were higher in responders than in non-responders, in early on-treatment samples of patients with melanoma treated with neoadjuvant ICI [169]. These findings were also supported by recent analyses of a TCGA cohort that demonstrate an association between a plasmablast-like B cell signature and improved survival, as well as with an increased expression of CD8a and infiltration of CD8^+^ T cells [170].

Recently reported meta-analysis on whole-exome and transcriptomic data of ICI-sensitive tumors, identified CCR5 and CXCL13 as T-cell-intrinsic markers of ICI sensitivity. CXC chemokine ligand 13 (CXCL13) exclusively binds CXC chemokine receptor type 5 (CXCR5), which plays a critical role in immune cell recruitment and activation and regulation of the adaptive immune response. CXCL13 is a key molecular determinant of the formation of TLSs [173]. Selective expression of CCR5 and CXCL13 in neoantigen-specific T lymphocytes suggests that a key feature of ICI-responsiveness is the ability to sustain ongoing priming and recruitment of tumor reactive T cells supported by CXCR5^+^ lymphocytes, including both T and B cells, and form TLSs [169].

With regards to genomic biomarkers, signatures encompassing T cell-specific gene expression profiles (GEP) have attracted a lot of interest. A 28-gene set including genes enriched for cytolytic activity (e.g., granzyme A/B/K, PRF1), cytokines/chemokines secretion (CXCR6, CXCL9, CCL5, and CCR5), T cell markers (CD3D, CD3E, CD2, IL2RG [encoding IL-2Rγ]), NK cell activity (NKG7, HLA-E), antigen presentation (CIITA, HLA-DRA), and additional immunomodulatory factors (LAG-3, IDO1, SLAMF6). A signature including all the above was shown to predict response to anti–PD-1–directed therapy in melanoma patients, and it appears to be a promising multi-dimensional biomarker, currently undergoing validation in larger clinical trials [152].

Dysfunction of antigen-presenting system was confirmed as one of the mechanisms of resistance to CPI, mostly related to HLA class I loss [174]. Β-microglobulin 2 gene (B2M) plays an important role in an assembly of all HLA class I complexes, as well as antigen transport [175]. Mutations in B2M are commonly associated with complete or partial loss of HLA class I function. Alterations in this gene are commonly identified in non-responders, and much less are seen in responders [176]. Accordingly, normal expression of B2M correlates with better overall survival of melanoma patients, treated with immunotherapy [177].

Clonality and diversity of intratumoral and peripheral TCR repertoire have also recently emerged as potential markers of responsiveness to immune checkpoint inhibitors in multiple cancer settings including melanoma [178,179,180,181,182,183,184,185]. Overall, these studies suggest that patients who exhibit a higher diversity of TCR repertoire before treatment and early evolution of the T cell populations after treatment start have a greater chance to benefit from immunotherapy. An association between high pre-treatment clonality of the T cell population and the probability of response to anti-PD-1 antibodies has also been observed [178]. Interestingly the same parameter showed an association with a lower chance of response to anti-CTLA-4 agents in melanoma [178]. This suggests that in the presence of an already expanded and ineffective clone of tumor-specific T cells, the use of anti-CTLA-4 antibodies will not be able to overcome cancer immune escape.

Asides from the intrinsic immune pathways directly connected to shaping the immune response, accumulating evidence underlines the presence of extrinsic factors also playing a pivotal role. Extracellular matrix (ECM) is emerging as another key element that can influence response to anticancer drugs, including immunotherapeutic agents [186]. ECM has indeed been demonstrated to regulate immune cell trafficking and activation, as well as cancer-associated antigen release and presentation [186]. High collagen levels have been associated with CD8^+^ suppression and exhaustion and with a lower probability of deriving benefit from anti-PD-1/PD-L1 blockade in melanoma and lung cancer [179,187]. Collagen is known to exert an inhibitory effect on T cells through the *SHP1* pathway activation, via the leukocyte-specific collagen receptor *LAIR1* [188]. These observations indicate ECM remodeling as a potential strategy to overcome resistance to immunotherapy. As an example, Peng et al. demonstrated the possibility to restore sensitivity to anti-PD-1 antibodies by inhibiting intratumoral collagen deposition or blocking the *LAIR1-SHP1* pathway [187].

*The gut microbiome as a biomarker in melanoma.* The gut microbiota influence metabolism, antitumor activity, and toxicity profile of a broad variety of anticancer treatments, including immunotherapy [189,190,191,192,193]. An association between higher microbiota biodiversity and a greater probability to benefit from anti-PD-1/PD-L1 antibodies has been reported for a variety of cancer types, including melanoma [194]. The abundance of specific bacteria populations such as *Bifidobacterium* and *Ruminococcaceae* has been described in the gut microbiota of patients benefitting from immunotherapy [194,195]. On the other hand, the microbiota of non-responders appears to be enriched with different strains such as *Bacteroidaes* and *Clostridium* species [194,196]. Several studies indicate a correlation between antibiotic-induced gut microbiota dysbiosis and poor clinical outcomes in cancer patients receiving immunotherapy [196,197,198,199,200].

The mechanisms underlying the relationship between microbiota composition and response to immune checkpoint inhibitors are not fully understood. Preclinical evidence indicates that microbiota can induce and activate multiple immune effectors, including NK, dendritic, CD4^+^, and CD8^+^ T cells, as well as Tregs and monocytes, through the production of different metabolites such as short-chain fatty acids, inosine, and bile-acid [201,202,203,204,205,206,207,208]. Interestingly, gut microbiota not only can influence the immune cell population but may also have a role in the remodeling of the extracellular matrix [209]. For example, adhesin A secreted by *Fusobacterium* species, *Bacteroides fragilis* toxin, and gelatinase E produced by *E faecalis* can modulate cell-to-cell interaction and mediate tumor progression and metastatic spread [210]. On the other hand, cross-reactivity between cancer cells and microbiota antigens can augment cancer cell immunogenicity and hence, promote ICI efficacy [211,212].

Following these findings, significant efforts have been made to develop effective strategies to correct gut dysbiosis with an aim to prevent or revert resistance to immunotherapy, with the strongest evidence supporting for fecal microbiota transplantation (FMT) so far [202]. The potential role of FMT in overcoming immunotherapy resistance is supported by preclinical and early clinical data. FMT from patients with cancer who responded to immunotherapy demonstrated an increased probability of benefit from anti-PD-1 blockade in xenograft mice [213]. In these models *A. muciniphila* and *E. hirae* shown the ability to induce secretion of IL-12 from dendritic cells, supporting immune-surveillance. Oral administration of *A. muciniphila* demonstrated the capability to revert resistance to immune checkpoint inhibitors in xenografts receiving FMT from non-responders [213]. Two proof-of-concept studies tested the efficacy of FMT administered by colonoscopy as a potential strategy to revert anti-PD-1 resistance in melanoma patients with disease progression following anti-PD-1 treatment [214,215]. FMT resulted in prolonged modification of the microbiota of the recipients and an objective response or prolonged disease stability lasting for more than 12 months was observed in 36% of the patients enrolled in the two studies. In patients who responded to ICI, FMT induced changes in the cytokine profile, with down-regulation of CCL2, IL-8, IL-18, IL12p70, and IFNγ and upregulation of IL-21, CXCL13, IL-5, IL-13, IL-10, TNF, and TRAIL [215]. Unsupervised single cell analysis of blood samples collected at baseline ad at serial time points during treatment was performed in one of the two studies [215]. This analysis demonstrated a higher prevalence of activated CD8^+^ cells in post-treatment samples from responding patients, characterized by overexpression of TIGIT, Tbet, and LAG-3 and suppression of CD27, on their cell surface. Moreover, responders had a higher percentage of peripheral T memory cells. Single-cell RNA sequencing of tissue samples collected before and after treatment was also extremely informative. This evaluation showed a higher prevalence of myeloid cells and Tregs in post-treatment biopsies from non-responders, whilst an increased expression of MHC class II genes was observed in samples obtained by patients who responded to treatment. Notwithstanding these encouraging results, FMT is an invasive procedure and therefore alternative approaches to modulate microbiome in cancer patients are currently under investigation including the use of probiotics, dietary intervention, and engineered microbiome (NCT04753359, NCT03637803, NCT03686202) [202].

## 4. Biomarkers of Disease Activity/Treatment Monitoring in Melanoma

*ctDNA as a tool.* Circulating tumor DNA (ctDNA) is the component of fragmented cell-free DNA (cfDNA) derived from tumor cells. Genomic profiles based on ctDNA are highly concordant with those of primary tumors, and ctDNA levels correlate directly with tumor burden. The detection of ctDNA following radical surgery may identify patients at the highest risk of clinical relapse and provide a “real-time” disease status at the molecular level. Therefore, the role of ctDNA as a non-invasive biomarker of molecular residual disease is widely evaluated in multiple tumor types. Liquid biopsy techniques measuring ctDNA can identify cancer DNA traces at the microscopic level [216,217]. This approach offers the possibility to monitor disease longitudinally and enables timely detection of disease progression or relapse [218].

Quantitative and digital polymerase chain reactions (like digital droplet PCR, ddPCR) were the first ctDNA detection approaches for selective gene targets [219,220]. Although highly validated, ddPCR is limited in its breadth to cover a large number of mutations in a tumor-specific and personalized way for each patient. Application of next-generation sequencing-based techniques is increasingly growing and allows for the detection of multiple mutations per patient and high sensitivity ctDNA tracking [221,222]. Next-generation sequencing (NGS) methods applied in a “tumor-informed” manner (i.e., tissue analysis is conducted to verify tumor specificity of aberrations observed in plasma) have the advantage of providing a signature of putative cancer-derived aberrations through analysis of concordant mutations [223,224]. Highly sensitive assays have permitted novel tracking of dynamic changes during therapy in a more sensitive way compared to ddPCR. For example, in early lung, colorectal, and breast cancers, postoperative detection of ctDNA prognosticates disease recurrence with high accuracy [225,226,227,228,229,230].

*ctDNA as disease predictor in melanoma.* Independently of standard AJCC staging, pre-operative detection of ctDNA in patients with stage III melanoma is associated with a 3-fold increased likelihood of disease relapse and half the time-to-distant metastatic relapse (6.2 months vs. 13.9 months (HR 1.59; 95% CI 1.0–2.52; *p* = 0.027)) [231,232]. This translates into a significantly shorter median melanoma-specific survival of 17.6 months compared with 49.4 months in patients with undetectable ctDNA levels (HR 2.11; 95% CI 1.20–3.71, *p* < 0.01), confirming the validity of ctDNA as a prognostic biomarker in early-stage melanoma. With serial assessments of ctDNA, a retrospective study of patients with stage III melanoma and no radiological evidence of disease, demonstrated that detection of *BRAF/NRAS* mutant ctDNA (by digital droplet PCR, ddPCR) at a single timepoint within 12 weeks of surgery was associated with worse disease-free interval and OS [233].

The detectability of ctDNA after surgical resection in melanoma patients is low, with detection rates below 25% even when using highly sensitive NGS-based assays [218,233,234,235,236]. Significant correlation has been observed between the presence of detectable ctDNA after surgery and shorter survival outcomes [232,233,234,237]. Despite the limited amount of data some studies seem to indicate that the administration of adjuvant immunotherapy might prolong the progression free survival of patients with positive ctDNA after surgery, nullifying the difference with the ctDNA negative population [232,234].

Perhaps the most robust research on the predictive value of ctDNA in melanoma, both in terms of patient sample size and use of most advanced ctDNA sequencing methodology so far, was conducted and reported as part of the CheckMate 915 study translational analysis. Long et al. confirmed in a cohort of 1127 patients with either resected stage IIIB-D and IV melanoma, that ctDNA detection at baseline can be an efficient predictor of recurrence-free survival for patients undergoing ICI treatment [237]. The predictive value was even more robust when ctDNA detection was combined with tumor TMB level and a tumor-derived IFNγ signature including HLA-DRA, CXCL9, GZMA, PRF1, CCR5, IFNG, CXCL10, IDO1, STAT1, and CXCL11 [152]. Overall, current evidence supports the use of postoperative ctDNA measurement to enable risk stratification of disease recurrence.

*ctDNA and immunotherapy response in melanoma.* Several studies have also investigated the potential use of ctDNA to predict anticancer therapy efficacy in melanoma patients [238,239]. These studies might be heterogeneous in terms of patient population and methodology used for ctDNA detection; however, they offer some important insights. Baseline levels and dynamic changes of ctDNA quantified with digital droplet PCR or with NGS-based assays were demonstrated to predict survival outcomes and response to immunotherapy in patients with metastatic disease [134,240,241,242]. Interestingly Lee et al. observed that ctDNA quantification might also be a valuable tool to distinguish patients who are having pseudo-progression on ICI from those whose disease is truly progressing [243].

Aside from the quantification of tumor burden and minimal residual disease, ctDNA evaluation can provide important information regarding the molecular features of cancer cells. The use of blood-based assays to evaluate known predictors of immunotherapy sensitivity, such as tumor mutational burden and microsatellite instability is rapidly expanding [244,245,246,247,248,249]. This approach not only overtakes the need for tissue samples but might also overcome the issue of tumor heterogeneity. Furthermore, ctDNA analysis is a promising methodology to study clonal evolution and the mechanisms underlying drug resistance [218,250,251]. Li et al. performed ctDNA analysis in 12 patients with NSCLC treated with pembrolizumab [252]. Serial samples were collected before and during treatment and sequenced against a 329 pan-cancer gene panel. Increasing levels of a *PTCH1* mutation were observed in a patient who was developing a new metastatic lesion more than 3 months before disease progression in CT scans. The authors also observed the sequential emergence of two different acquired mutations of *B2M* in another patient whose disease progressed after achieving an initial response to pembrolizumab. Similarly, Jin et al. used a 425-genes next-generation sequencing panel to profile tumor tissue and blood samples collected before and after treatment in 46 gastric cancer patients treated with anti-PD-1 antibodies [253]. Baseline *TGFBR2*, *RHOA*, and *PREX2* mutations were identified as predictors of shorter PFS. To investigate potential markers of acquired resistance, blood samples obtained at the time of ICI-resistance emergence from 16 patients were analyzed. *FOXL2* gene mutations and copy number variations of *FGFR2* were identified as new alterations in two patients and a new *RHOA* mutation was observed in another patient suggesting possible implication of these genes in driving the newly acquired immune treatment resistance.

Evidence supporting the use of ctDNA to study clonal dynamics exists in melanoma too. Takai et al. reported the results of whole exome sequencing of sequential ctDNA samples collected from 14 patients with metastatic melanoma receiving ICI [254]. Newly emerged ctDNA mutations—such as in *ARID1B*—were observed at the timepoint of disease progression. Overall, this data showcase ctDNA as a powerful tool to guide treatment decision and inform the design of clinical trials testing novel therapeutic strategies to overcome drug resistance. In a disease like melanoma, where ICI-resistance therapeutic strategies are critically needed, ctDNA can be proven immensely instrumental.

## 5. Molecular Biomarkers in Non-Melanoma Skin Cancers

Cutaneous squamous cell carcinoma (cSCC) is affecting more than 1 million people in the United States, every year. Incidence is rising due to increasing sun exposure and an aging population. Approximately 2–5% of the patients present with locoregionally advanced cSCC [255]. Ultraviolet exposure, with subsequent DNA damage that leads to a high tumor mutational burden (TMB), as well as a state of immunosuppression, is the main risk factors for cSCC development and interestingly, both factors correlate with superior response to immunotherapy [116,256]. In the recent decade, multiple trials confirmed the efficacy of ICI in various stages of the disease [117,118,124,257,258,259]. In most published studies, tumor response and survival subgroup analyses were mainly based on clinical characteristics, along with well-established biomarkers, such as TMB and PD-L1 status. PD-L1 positivity (>1%) is reported to correlate with better response to treatment, although up to 20% response rate was observed in PD-L1 negative (less than 1%) tumors [124,260,261]. Tumor mutations burden is considered to be generally high in cSCC, but nevertheless, this can vary significantly. In a recently published study with neo-adjuvant cemiplimab [124], patients’ median tumor mutational burden of 61.1 mut/Mb was found to correlate with the achievement of major or complete pathologic response [124]. From an immune microenvironment perspective, as expected, infiltration of T cells, including CD8^+^ T cells and CD4^+^ Th1 cells was associated with superior treatment response. This was accompanied by a significantly higher expression of IFNγ-related immune genes. On the other hand, abundant Tregs and CD68^+^ myeloid cells expressing the inhibitory checkpoint V-domain immunoglobulin suppressor of T cell activation (VISTA), were reported as biomarkers of resistance [261].

Merkel cell carcinoma (MCC) remains a very rare cancer type, usually affecting elderly patients, with 5-year overall survival ranging from 63% in stage I, to 13% in stage IV [125]. Two main factors are implicated in MCC tumorigenesis: Merkel cell polyomavirus (MCPyV) or chronic UV exposure. MCC is highly immunogenic due to the presence of either MCPyV-derived viral antigens or UV exposure-associated neoantigens, thus representing a disease amenable to ICI treatment [262,263,264,265]. Mutational profile varies between these two subgroups. For MCPyV-negative patients, next-generation sequencing (NGS) and whole exome sequencing (WES) have revealed a high tumor mutational burden (TMB), with recurrent mutations in *TP53* and *RB1*. On the other hand, MCPyV-positive patients demonstrated 100-fold lower TMB than MCPyV-negative tumors, without a pattern of recurrent mutations [263,264]. Notwithstanding these differences, both subtypes remain highly immunogenic and responsive to ICI, independent of MCPyV or TMB status. As a result, to this point, TMB cannot be utilized as a predictive biomarker for response to ICI [114,266,267]. In the same vein, PD-L1 does not represent a robust biomarker either for this type of malignancy [266,267].

On the other hand, several studies have reported clinical biomarkers as surrogates of improved response to ICI. These include prior lines of treatment, where the use of ICI as a first-line therapy was associated with two-fold response rates as compared to administering it in the second-line setting [120,121,126,266]. Moreover, other clinical features associated with prolonged survival are less advanced disease at diagnosis, significant tumor shrinkage after exposure to ICI, limited number of tumor-affected organs, shorter disease-free interval between completion of initial treatment and recurrence and lack of immunosuppression [114,127,268]. Evaluation of peripheral blood indices showed longitudinal correlation of lower neutrophil to leukocyte ratio with improved patient outcomes. However, the correlation was not confirmed when the neutrophil/leucocyte ratio was assessed at baseline or individual timepoints only during treatment with ICI [127]. From immunological and molecular biomarker perspective, predominance of CD8^+^ T cells among tumor infiltrating lymphocytes (TIL) at baseline and presence of *ARID2* and *NTRK1* mutations were associated with a favorable response to PD-1/PD-L1 ICI therapy in advanced MCC [114,120,268].

Lastly, basal cell carcinoma (BCC) is the most common type of malignancy, although it rarely evolves into advanced disease, due to very latent behavior [255]. Along with cSCC, BCC has one of the highest mutational burdens among cancer types [269]. Due to the high efficacy of hedgehog inhibitors (HHI) in advanced BCC, immunotherapy was not evaluated in this disease only until recently [270]. After initial positive results of ICI following progression on HHI [122], multiple trials were opened to evaluate the role of ICI in the BCC treatment algorithm (Table 3). From a biomarker perspective, very limited data is currently available, although exploratory biomarker analysis from a recently reported trial [122] showed no clinically meaningful associations between objective response and any of the biomarkers, established as predictive in other tumor types, including PD-L1 expression, TMB or MHC-I expression.

## 6. Conclusions

The unparalleled survival improvement brought about by immune checkpoint treatment in patients with melanoma has undoubtedly changed the therapeutic landscape of the entire field of oncology, as well as the current focus of cancer research. One could draw similarities to the discovery of intracellular pathways and the advent of targeted treatments which was yet another important milestone for cancer clinical practice at the end of the previous century. In contrast to intracellular pathway blockade though, immune checkpoint inhibition offers longevity of disease response and the so-elusive prospect of a cure in the metastatic setting.

Notwithstanding the success of immunotherapy, there is still a significant number of patients with melanoma who have either primary or acquired resistance to immune checkpoint blockade. This particular cohort of patients is severely disadvantaged by the paucity of clinically validated biomarkers of immunotherapy resistance and in the worst-case scenario, they also have to endure the immune-related adverse events that come with treatment. There is an imperative need to discover the biological mechanisms underpinning this resistance and to determine high-sensitivity molecular markers to describe these mechanisms.

The focus now is turning to unraveling immune pathways that work synergistically with the known immune targets, with an aim to push the boundaries of treatment resistance even further. These discoveries will not be fruitful though unless accompanied by biomarkers of response and resistance that will guide target selection and treatment sequencing. The moment when “personalized immunotherapy” becomes a reality in clinical practice will hopefully come in the not-too-distant future.

## Figures and Tables

**Figure 1 ijms-24-01294-f001:**
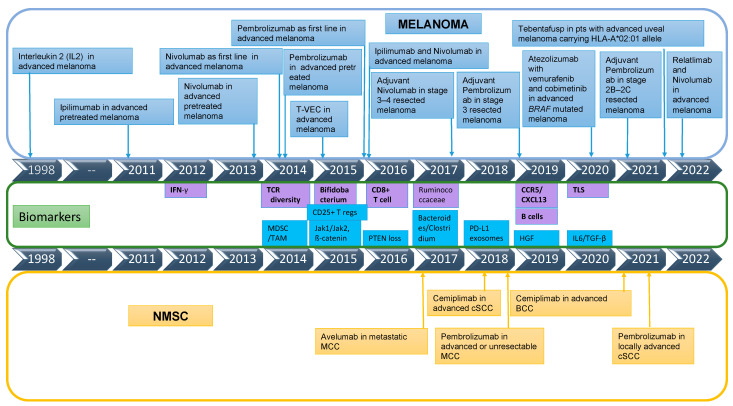
Timeline of immunotherapy agents’ approval for melanoma and non-melanoma skin cancer treatment. NMSC—non-melanoma skin cancer, cSCC—cutaneous squamous cell carcinoma, BCC—basal cell carcinoma, MCC—Merkel cell carcinoma, T-VEC—Talimogene laherparepvec; IFN-γ—interferon gamma; TCR—T cell receptor; CCR5—chemokine receptor 5; CXCL13—chemokine ligand 13; TLS—tertiary lymphoid structures; TAM—tumor-associated macrophages; MDSC—myeloid-derived suppressor cells; T regs—regulatory T cells; HGF—hepatocyte growth factor; IL6—interleukin 6; TGF-β—Transforming growth factor beta; purple colour for biomarkers: biomarkers of response; blue colour for biomarkers: biomarkers of resistance.

**Figure 2 ijms-24-01294-f002:**
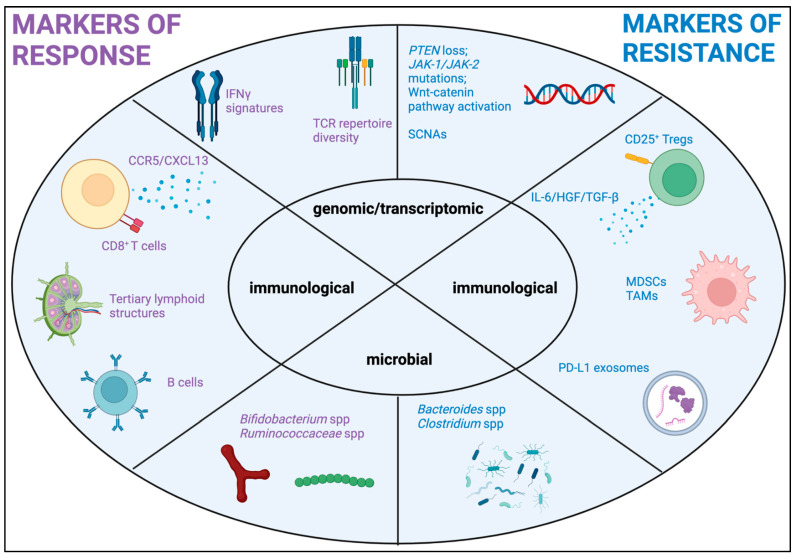
Biomarkers of response/resistance to immunotherapy in melanoma. purple color: markers of response; blue color: markers of resistance.

**Table 1 ijms-24-01294-t001:** Ongoing clinical trials with immunotherapy in advanced cutaneous melanoma (as assessed on 13 November 2022). * Denotes studies that also include patients with tumor types other than melanoma.

Type of Intervention	Targets	Agents	Number of Patients	Phase	NCT Number
**Novel Immune Pathways**					
	Anti-PD-1 plus anti-TIGIT (IV) or anti-PD-1 + Coxsackievirus (IT) or anti-PD-1 + ILT-4 or anti-PD-1/anti-LAG-3 or anti-PD-1 + retinoid	Pembrolizumab + vibostolimab or pembrolizumab + gebasaxturev or pembrolizumab + MK-4830 or favezelimab or pembrolizumab + all-trans retinoic acid (ATRA)	90	1/2	NCT04303169
	Anti-CTLA-4/anti-PD-1 +/− anti-interleukin 8 antibody	Ipilimumab/Nivolumab +/− BMS-986253 (HuMax-IL8)	372	1/2	NCT03400332
	IL-2Rβ/γ-selective IL-2 variant +/− anti-PD-1 +/− TLR7/8 agonist	Pembrolizumab +/− TransCon IL-2 β/γ +/− TransCon TLR7/8 agonist	317 *	1/2	NCT05081609
	IL-2 superkine +/− ICI	MDNA11 +/− ICI	100 *	1/2	NCT05086692
	T cell engager +/− anti-PD-1 +/− anti-CTLA-4	Tebentafusp +/− durvalumab +/− tremelimumab	312	1/2	NCT02535078
	IL-12 Fc-fusion protein +/− anti-PD-1	BMS-986415 +/− Nivolumab	473 *	1/2	NCT04423029
	CD40 agonist +/− anti-PD-1	APX005M + pembrolizumab	41	1/2	NCT02706353
	CXCR1/2 inhibitor + anti-PD-1	SX-682 + pembrolizumab	77	1	NCT03161431
	STING agonist + anti-PD-L1	SB 11285 + atezolizumab	110 *	1	NCT04096638
	Anti-TIGIT + anti-PD-1	AB154 + AB122	26	2	NCT05130177
	Interleukin (IL-)15 and IL-15 receptor alpha + anti-PD-1	NIZ985 +/− spartalizumab	110 *	1	NCT04261439
	mAb Specific to B-and T-Lymphocyte Attenuator (BTLA)	JS004	156 *	1	NCT04773951
	PD-L1/IDO peptide vaccine + anti-PD-1	PD-L1/IDO peptide vaccine + nivolumab	50	1/2	NCT03047928
	Anti-CCR8 +/− anti-PD-1	BAY3375968	270 *	1	NCT05537740
	TGFβ1 inhibitor +/− anti-PD-L1	SRK-181 +/− anti-PD-L1	200 *	1	NCT04291079
	Anti-PD-1 inhibitor/OX40 agonist	SL-279252 (PD1-Fc-OX40L)	87 *	1	NCT03894618
	Granulocyte Macrophage-colony stimulating factor + anti-PD-1	Sargramostim + pembrolizumab	30	2	NCT04703426
**Adoptive cell therapy**					
	PD-1 knock out TILs	IOV-4001	53 *	1/2	NCT05361174
	Adoptive autologous cellular therapy + HDAC + DNMT inhibitors	MAGE-C2/HLA-A2 TCR T cells + azacytidine + valproic acid	20	1/2	NCT04729543
	CD20 CAR transduced T cells	MB-CART20.1	15	1	NCT03893019
	Autologous alpha-type-1 polarized dendritic cells (alphaDC1)/TBVA vaccine + cytokine modulating regimen	TBVA vaccine intatolimod + IFN-alpha2b + celecoxib	24	2	NCT04093323
	Autologous neoantigen-specific T-cell product	NEO-PTC-01	52	1	NCT04625205
**Oncolytic viral therapy**					
	Rhinovirus targeting poliovirus receptor CD155 (IT) +/− anti-PD-1	Lerapolturev (IT) +/− anti-PD-1	56	2	NCT04577807
	Oncolytic vesicular stomatitis virus expressing human IFNβ (IT or IV) +/− anti-PD-1 +/− anti-CTLA-4	Voyager V1 +/− cemiplimab +/− ipilimumab	152 *	2	NCT04291105
	Anti-PD-1 + oncolytic HSV type virus expressing anti-PD-1 antibody and IL-12 (IT)	MVR-T3011 (IT) +/− pembrolizumab	10	1/2	NCT04370587
	Herpes simplex virus type 2 strain HG52 +/− anti-PD-1	OH2 +/− nivolumab	30 *	1/2	NCT04386967
	Adenovirus with TMA-CD40L and 4-1BBL transgenes + anti-PD-L1	LOAd703 + atezolizumab	35	1/2	NCT04123470
	Great ape Adenoviral (GAd)/Modified Vaccinia Ankara (MVA) boosts with personalised patient neoantigens + anti-PD-1	NOUS-PEV + pembrolizumab	34 *	1	NCT04990479
	Anti-PD-1 + oncolytic HSV type 1 virus (IT)	Pembrolizumab + RP1	300 *	2	NCT03767348
	Anti-PD-1 plus vaccinia virus encoding transgenes for Flt3 ligand, anti-CTLA-4 antibody and IL-12 cytokine, (IT or IV)	Pembrolizumab + TBio-6517 (IV or IT)	138 *	1/2	NCT04301011
**Targeted treatment plus immunotherapy**					
	PI3K inhibitor + anti-PD-1	Duvelisib + nivolumab	42	1/2	NCT04688658
	ETBR inhibitor +/− anti-PD-1	ENB003 +/− pembrolizumab	130	1/2	NCT04205227
	Anti-VEGF + anti-PD-L1	Bevacizumab + atezolizumab	30	2	NCT04356729
	Integrin alpha-V/beta-8 Antagonist	PF-06940434	122	1	NCT04152018
	p38 inhibitor + anti-PD-1 or anti-CTLA-4/anti-PD-1	ARRY-614 + nivolumab or nivolumab/ipilimumab	144 *	1/2	NCT04074967
	Hedgehog pathway inhibitor + anti-PD-1	Sonidegib + Pembrolizumab	45 *	1	NCT04007744
	Wnt/β-catenin inhibitor +/− anti-PD-1	E7386 +/− pembrolizumab	108 *	1/2	NCT05091346
	PARP inhibition + anti-PD-1	Olaparib + pembrolizumab	41	2	NCT04633902
	“Velcrin” that triggers the formation of a complex of two proteins called SLFN12 and PDE3A	BAY2666605	89	1	NCT04809805
	Heparanase inhibitor (TAMs inhibitor) + anti-PD-1 + metronomic chemotherapy	Pixatimod (PG545) + Nivolumab + cyclophosphamide	61	2	NCT05061017
	Anti-CTLA-4/anti-PD-1 +/− glutamate modulator	Ipilimumab/Nivolumab +/− troriluzole	108	2	NCT04899921
	MDM2 inhibitor + anti-PD-1	APG-115 + pembrolizumab	224	1/2	NCT03611868
	Multi-tyrosine kinase inhibitor + anti-PD-1	Cabozantinib + pembrolizumab	39	1/2	NCT03957551
	BTK inhibitor + anti-PD-1	Ibrutinib + pembrolizumab	23	1	NCT03021460
	ATR inhibitor +/− anti-PD-L1	Ceralasertib +/− durvalumab	195	2	NCT05061134
	Anti-PD-1 plus NLRP3 inhibitor	Pembrolizumab + Dapansutrile	26	1/2	NCT04971499
**Vaccine-based treatment**					
	Melanoma HLA-restricted peptides vaccine +/− anti-CD27	6MHP +/− CDX-1127	30	1/2	NCT03617328
	DNA vaccine encoding Tyrosinase-Related Protein 2 (TRP2) and gp100	SCIB1 DNA vaccine	87	2	NCT04079166
	mRNA-encoded with tumor-associated antigens vaccine + anti-PD-1	BNT111 + cemiplimab	180	2	NCT04526899
**Antibody drug conjugates**					
	ADC against CD228	SGN-CD228A	275 *	1	NCT05571839
	ROR2-targeted antibody drug conjugate (CAB-ROR2-ADC) +/− anti-PD-1	BA3021 +/− pembrolizumab	420 *	2	NCT03504488

**Table 2 ijms-24-01294-t002:** Ongoing clinical trial testing immunotherapy strategies in patients with uveal and mucosal melanoma (as assessed on 13 November 2022).

Target Population		Type of Treatment	Intervention	Number of Patients	Phase	NCT Number
**Uveal Melanoma**		Local treatment + anti-PD-1 and anti CTLA-4	Yttrium 90 + Ipilimumab + Nivolumab	26	1/2	NCT02913417
			Melphalan percutaneous hepatic perfusion + Ipilimumab + Nivolumab	83	1/2	NCT04283890
			Immunoembolization +IPI/NIVO	14	2	NCT03472586
			Ipilimumab + Nivolumab + Tumor Treating Fields	10	1	NCT05004025
			Stereotactic RT + Ipilimumab + Nivolumab	40	2	NCT05077280
		Anti-PD-1 + Anti-LAG-3	Nivolumab + Relatlimab	27	2	NCT04552223
		Anti-PD-1 and anti CTLA-4 + TLR9 agonists	Nivolumab + Ipilimumab + intrahepatic SD-101	80	1	NCT04935229
		Anti-PD-1 and anti CTLA-4 + anti-arginine	ADI-PEG20 + Ipilimumab + Nivolumab	9	1	NCT03922880
		Anti-CTLA-4 + cell therapy	CD8+ SLC45A2-Specific T Lymphocytes + Cyclophosphamide, Aldesleukin, and Ipilimumab	30	1	NCT03068624
		Cell therapy	Tumor Infiltrating Lymphocytes	47	2	NCT03467516
				10	1	NCT05607095
			Dendritic cells + tumor RNA	200	3	NCT01983748
		Anti-PD-1 + epigenetic modulators	Pembrolizimab + Etinostat	29	2	NCT02697630
		Anti-PD-1 + Anti-VEGF	Pembrolizumab + Lenvatinib	54	2	NCT05282901
				30	2	NCT05308901
		Anti-PD-1 + Anti-DDR	Pembrolizumab + Olaparib	37	2	NCT 05524935
**Mucosal Melanoma**	Anti-PD-1 + local treatment		Adjuvant Pembrolizumab + hypofractionated RT	16	2	NCT04318717
	Anti-PD-1 + chemotherapy + local treatment		Adjuvant Toripalimab + chemotherapy or RT	45	2	NCT04879654
	Anti-PD-1 + Anti-VEGF		Neoadjuvant Pembrolizumab + Lenvatinib	44	2	NCT05545969
				26	2	NCT04622566
			SHR-1210 + Apatinib	40	2	NCT03986515
			Adjuvant Nivolumab +/− Cabozantinib	99	2	NCT05111574
			Nivolumab + Axitinib	20	2	NCT05384496
			Toripalib + Axitinib	30	2	NCT04180995
				99	2	NCT03941795
	Anti-PD-1 + anti-TGFβ		Pembrolizumab + Vactosertib	30	2	NCT05436990
	Anti-PD-1 + anti CD40 + chemotherapy		YH003 + Pembrolizumab + Paclitaxel	43	2	NCT05420324
	Anti-PD-1 + chemotherapy		Nivolumab + Decitabine/Cedazuridine	30	1/2	NCT05089370
	IL-2 agonists		ALKS 4230	110	2	NCT04830124

**Table 3 ijms-24-01294-t003:** Ongoing clinical trials with different types of ICI in NMSC (as assessed on 13 November 2022).

Study Title	Tumor Type	Intervention	Phase	NCT Number
Neo-adjuvant Nivolumab or Nivolumab With Ipilimumab in Advanced Cutaneous Squamous Cell Carcinoma Prior to Surgery	cSCC	Ipilimumab Nivolumab	2	NCT04620200
Neoadjuvant Plus Adjuvant Treatment With Cemiplimab in Cutaneous Squamous Cell Carcinoma	cSCC	Cemiplimab	2	NCT04632433
Neoadjuvant Pembrolizumab in Cutaneous Squamous Cell Carcinoma	cSCC	Pembrolizumab	2	NCT05025813
Atezolizumab Before Surgery for the Treatment of Regionally Metastatic Head and Neck Squamous Cell Cancer With an Unknown or Historic Primary Site	cSCC	Atezolizumab	2	NCT05110781
Avelumab With or Without Cetuximab in Treating Patients With Advanced Skin Squamous Cell Cancer	cSCC	Avelumab Cetuximab	2	NCT03944941
Phase II Study of Peptide Receptor Radionuclide Therapy in Combination With Immunotherapy for Patients With Merkel Cell Cancer	MCC	Pembrolizumab, Lutetium Lu 177 dotatate	2	NCT05583708
Neoadjuvant Lenvatinib Plus Pembrolizumab in Merkel Cell Carcinoma	MCC	Pembrolizumab, Lenvatinib	2	NCT04869137
Navtemadlin (KRT-232) With or Without Anti-PD-1/Anti-PD-L1 for the Treatment of Patients With Merkel Cell Carcinoma	MCC	KRT-232, Avelumab	2	NCT03787602
Adjuvant Avelumab in Merkel Cell Cancer	MCC	Avelumab, placebo	2	NCT03271372
Testing Pembrolizumab Versus Observation in Patients With Merkel Cell Carcinoma After Surgery, STAMP Study	MCC	Pembrolizumab, placebo	3	NCT03712605
Intralesional Cemiplimab for Patients With Cutaneous Squamous Cell Carcinoma or Basal Cell Carcinoma	cSCC, BCC	Cemiplimab intralesional	1	NCT03889912
Nivolumab Alone or Plus Relatlimab or Ipilimumab for Patients With Locally-Advanced Unresectable or Metastatic Basal Cell Carcinoma	BCC	Nivolumab, Relatlimab	2	NCT03521830
Neoadjuvant-Adjuvant Pembrolizumab in Resectable Advanced Basal Cell Carcinoma of H&N	BCC	Pembrolizumab	1	NCT04323202
Anti-PD1-antibody and Pulsed HHI for Advanced BCC	BCC	Cemiplimab, Sonidegib	2	NCT04679480

## Data Availability

Not applicable.

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
