# Peer review of "Shaping the Future of Immunotherapy Targets and Biomarkers in Melanoma and Non-Melanoma Cutaneous Cancers"

_ijms, 2023, doi:10.3390/ijms24021294_

Round 1

Reviewer 1 Report

In the manuscript titled ‘Shaping the future of immunotherapy targets and biomarkers in melanoma and non-melanoma cutaneous cancers’, the authors described the current status and future projections for the immunotherapy targets involved in melanoma and non-melanoma cutaneous cancers. The authors explained in-depth the standards and future prospects of the specific field of study. The manuscript is detailed and provides a rationale-oriented view of immunotherapy in melanoma and NMSC. The targets that have been approved and those in contention for approval have been described well for easy understanding for the reader. The data incorporation in figures for visualization adds to the aesthetic appeal of the manuscript. The timeline history gives an overview of the fast-paced research and its fruits in the form of approval. The conclusion showcases the importance of the evident rise of immunotherapy as not only a promising but useful approach with caution for further addition of potential sub-markers is also good. All in all, the manuscript is great and can be recommended for publication in the International Journal of Molecular Sciences. It was a good read and will be helpful for readers focusing on immunotherapy-related hypotheses in the context of melanoma and NMSC.

Author Response

We thank Reviewer 1 for their constructive feedback. Reviewer 1 has thoroughly described the objectives of this review and is suggesting publication within International Journal of Molecular Sciences. There were no particular points that needed addressing in the Review Report Form but nevertheless, we are extremely grateful for this feedback.

Sincerely yours,

Dr Anna Spreafico.

Reviewer 2 Report

The authors have presented a comprehensive review of the most recent advances in the immunotherapy of cutaneous melanoma and non-melanoma skin cancers discussing the achievements of multiple clinical trials and the role of important molecular biomarkers related to the treatment. This review will be of high interest to the readers of the journal.

Minor comments

1.     Line 445: the authors state “PD-L1 expression has not been proven to be a reliable biomarker in melanoma, as evidenced by exploratory analyses of large scale studies (130-132). “ The mentioned trials made a  conclusion that “The duration of response was sustained across stratification subgroups (according to BRAF mutation status, PD-L1 status, and metastasis stage) “. Besides, further in the manuscript, the authors discuss the role of PD-L1 such as:  “…the pre-treatment level of circulating exosomal PD-L1 was significantly higher in patients whose disease failed to respond treatment.” Line  512, or  “…PD-L1 positivity (> 1%) is reported to correlate with better response to treatment,….(124, 263, 264). “ Lines 747-748. Thus, could you, please, write the statement in Line 445 so that it doesn’t sound misleading?

2.     Line 258: write, please, the abbreviation “IO “ in full the first time mentioned.

Author Response

We sincerely thank Reviewer 2 for taking the time to read our manuscript and provide their valuable feedback. Below, we will provide a point-by-point response to Reviewer's 2 minor comments.

  1. Line 445: the authors state “PD-L1 expression has not been proven to be a reliable biomarker in melanoma, as evidenced by exploratory analyses of large scale studies (130-132). “ The mentioned trials made a  conclusion that “The duration of response was sustained across stratification subgroups (according to BRAF mutation status, PD-L1 status, and metastasis stage) “. Besides, further in the manuscript, the authors discuss the role of PD-L1 such as:  “…the pre-treatment level of circulating exosomal PD-L1 was significantly higher in patients whose disease failed to respond treatment.” Line  512, or  “…PD-L1 positivity (> 1%) is reported to correlate with better response to treatment,….(124, 263, 264). “ Lines 747-748. Thus, could you, please, write the statement in Line 445 so that it doesn’t sound misleading?

We sincerely thank the reviewer for pointing out that this may sound misleading for the reader.

Firstly, we would like to bring to the Reviewer's attention that although line 445 is referring to PD-L1 immunohistochemistry assay in metastatic melanoma, line 512-513 is referring to the quantification of PD-L1 exosomes by ELISA and not immunohistochemistry.

Moreover, in lines 747-748 (749 on the revised version attached), our statement on PD-L1 positivity refers to cutaneous squamous cell carcinoma and not melanoma. Therefore, statements on lines 512-513 and 747-749 are unrelated to melanoma and do not contradict what it is written on line 445, which refers to PD-L1 IHC in metastatic melanoma.

With regards to line 445, our statement highlights that PD-L1 positivity by immunohistochemistry, failed to stratify patients with metastatic melanoma into groups of patients that are highly likely to respond versus less likely to respond and therefore, can not be used as a predictive biomarker, specifically in metastatic melanoma.

This conclusion is supported by the fact that PD-L1 IHC is not used as predictive biomarker in clinical practice for metastatic melanoma, given that patients respond to treatment across different levels of PD-L1 expression.

We have altered our sentences on lines 445-446 and 513-514 (tracked changes), to make the above more clear for the reader.

  1. Line 258: write, please, the abbreviation “IO “ in full the first time mentioned.

Thank you for pointing out this omission in the text. We have added the explanation of "IO" (immuno-oncology) on line 258. 

We hope that the explanations above will be adequate for Reviewer 2 and the Editors to consider our manuscript for full publication.

Thank you,

Dr Anna Spreafico.
